# The Effects of Maternal Endocrinopathies and Exposure to Endocrine Disruptors During Pregnancy on the Fetus and Newborn

**DOI:** 10.3390/biomedicines13081965

**Published:** 2025-08-13

**Authors:** Ruth Fox, Su’ad Akinboro, Andrzej Kędzia, Elżbieta Niechciał

**Affiliations:** Department of Pediatric Diabetes, Clinical Auxology and Obesity, Poznan University of Medical Sciences, 60-572 Poznan, Poland; ruthie.fox@outlook.com (R.F.); akinboro@ualberta.ca (S.A.); akedzia@ump.edu.pl (A.K.)

**Keywords:** maternal endocrinopathies, diabetes, thyroid disease, adrenal disorders, polycystic ovary syndrome, endocrine disruptors, fetal development, infant

## Abstract

Maternal health has a profound impact on fetal development, influencing the risk of pediatric endocrine disorders both directly and indirectly through various biological and environmental mechanisms. Throughout pregnancy, several endocrine disorders can arise or be exacerbated due to the physiological changes that occur. An in-depth review of articles with evidence-based research discussing the significant effects of maternal endocrinopathies and endocrine disruptors on fetal development and infant health was conducted in this review paper. The most common endocrine disorder during pregnancy is gestational diabetes mellitus, which has an incidence rate of 2–16%, depending on ethnic origin. Maternal diabetes, apart from macrosomia and hypoglycemia, increases the risk for several pregnancy and neonatal complications such as stillbirth, perinatal mortality, and congenital malformations. Other endocrine issues occurring in pregnancy include alterations in thyroid hormone levels, obesity-related insulin resistance, Cushing syndrome, or polycystic ovarian syndrome, which all may negatively influence the fetus, as well as offspring development. Additionally, environmental exposure to harmful substances during pregnancy can disrupt endocrine function. Bisphenol A is the most common endocrine disruptor, which is particularly detrimental during gestation. Bisphenol A exposure is related to low birth weight, preterm birth, or developmental delays. Also, its exposition could be associated with an increased risk of obesity, metabolic disorders, and certain cancers later in life. Endocrinopathies and exposure to endocrine disruptors during pregnancy represent a challenging problem, being widespread and demanding appropriate management to reduce fetal and newborn complications.

## 1. Introduction

Maternal health plays a crucial role in fetal development, influencing the risk of pediatric endocrine disorders through direct physiological influences and indirect environmental factors [1,2]. Hormonal disorders and contact with endocrine-disrupting substances during pregnancy pose a complex and widespread health concern. The growing body of evidence from experimental and clinical studies suggests an interplay between endocrinopathies and endocrine disruptors, linking maternal health to fetal development and the subsequent risk of endocrine disorders [1,2,3,4,5,6,7,8,9,10,11,12].

The environment in which the fetus develops is a key factor in determining its outcomes, being crucial to its survival at birth and long-term physiology. Proper fetal growth results from various multidirectional interactions between the mother, placenta, and fetus. The maternal–endocrine–fetal unit serves as the interface for nutrient and gas exchange. Moreover, it is an essential source of protein and steroid hormones, and it is assumed to be a selective barrier, creating a protective intrauterine environment by reducing exposure of the fetus to external factors. Finally, the maternal–endocrine–fetal unit plays a role in controlling endocrine function and fetal growth by coordinating the production and secretion of hormones such as human chorionic gonadotropin (hCG), progesterone, estrogens, and placental lactogen (hPL). This unit is critical in regulating various physiological processes during pregnancy, influencing maternal health and fetal development [2].

Endocrinopathies, which are various conditions where the normal hormone regulatory, production, and supply function of the maternal endocrine system are disrupted, pose an increased risk of adverse fetal and child outcomes. This emphasis on reducing the unfavorable environment is further supported by research on epigenetics and alteration of gene expression, such as DNA methylation, which predisposes to the development of metabolic diseases later in adulthood [3,4,13,14]. The Developmental Origins of Health and Disease (DOHaD) theory established the association between the intrauterine environment during critical fetal developmental periods and long-term health outcomes. Numerous studies have replicated these findings regarding susceptibilities to disorders because of exposures to certain hormones and chemicals during gestation, thereby leading to programming of the fetal endocrine system [15]. These critical periods are known as windows of susceptibility and can manifest as turning points whereby metabolic disorders later in life occur because of the interaction between environmental influences and epigenetic adaptations [16].

In addition, environmental exposures to harmful substances during pregnancy can disrupt endocrine function. One of the most common endocrine disruptors is bisphenol A (BPA), a chemical frequently found in various plastics and resins. Exposure to BPA during critical periods of fetal development is associated with a range of negative consequences, including preterm birth, low birth weight, and other adverse pregnancy outcomes [17].

This narrative review investigates evidence on the impact of endocrinopathies and endocrine disruptors on fetal development. The paper aims to identify the most clinically relevant and current literature, thoroughly exploring how endocrine conditions and environmental factors affect fetal development. The analysis was not conducted through a systematic literature review by each author. The inclusion criteria were original research articles, including observational studies (cohort, case-control, cross-sectional); clinical trials; reviews; systematic reviews; and meta-analyses relevant to maternal endocrinopathies and endocrine disruptor exposure during pregnancy. Exclusion criteria included non-English language papers, studies with insufficient data, case reports, editorials, commentaries, non-peer-reviewed articles, duplicated articles, unavailable full texts, or abstract-only papers. The following electronic databases were searched for the most important full-text articles in English: PubMed, Google Scholar, EMBASE, Scopus, National Center for Biotechnology Information (NCBI), and Web of Science. All the articles published up to 2025 were reviewed for relevance with respect to the research question, abstract, and full text.

The search was done using the following keywords: endocrine disruptors; hormonal disruptors; endocrinopathies; fetal development; fetal growth; embryonic development; type 1 diabetes; type 2 diabetes; diabetes mellitus; gestational diabetes mellitus; maternal obesity; high body mass index (BMI); insulin resistance; epigenetics; thyroid disease; hyperthyroidism; thyrotoxicosis; hypothyroidism; Grave’s disease; Hashimoto thyroiditis; autoimmune hyperthyroidism; metformin; Cushing’s syndrome; hypercortisolism; polycystic ovarian syndrome; endocrine disrupting chemicals; folic acid; bisphenol A; phthalates; maternal nutrition; toxins; environmental factors; pregnancy complications; maternal–fetal interactions; placental interactions; obstetric complications; conceptional ages; interventions; preconception; and postpartum.

The search was conducted between October 2024 and June 2025. This review explores the critical impacts of maternal endocrinopathies such as diabetes, thyroid dysfunction, adrenal disorders, and polycystic ovary syndrome, as well as discusses various endocrine disruptors, focusing on bisphenol A and phthalate substances.

## 2. Obesity and Diabetes

Obesity, which is defined as a body mass index (BMI) ≥ 30 kg/m^2^ or higher, has been recognized as a medical crisis, with rates increasing globally, both in adults and children. In women of childbearing age, obesity is even more concerning, as the implications reveal a trend of adverse outcomes in fetal development [18].

Maternal overnutrition and obesity contribute to significant overall adverse effects on long-term health in progeny, including the development of obesity and other metabolic-related diseases. Evidence has advocated that excessive weight gain might have originated from environmental factors and maternal metabolic alterations at the early stages of fetal life. The link between maternal obesity and the higher risk of its development in offspring is not yet fully explored. However, studies on animal models shed light on the physiological mechanisms, suggesting that epigenetic modifications may be a key factor in disease susceptibility in the later stages of a child’s life. Environmental factors may cause epigenetic modifications by altering gene expression, mainly those responsible for cell survival and replication, without changing the deoxyribonucleic acid (DNA) sequence. The main epigenetic mechanisms include DNA methylation, histone modifications, and microRNA (miRNA); however, DNA methylation is hypothesized to be primarily related to obesity and diabetes. In DNA methylation, a methyl group is added to DNA on the C5 position of the cytosine ring, causing changes in the DNA’s appearance and structure. Recently, it has been shown that environmental stimuli, such as unhealthy dietary habits, mainly during pregnancy and early life, might initiate changes in DNA methylation, leading to hypermethylation or hypomethylation [19,20,21,22,23]. This relates to mechanisms causing permanent changes in the metabolic, physiologic, cellular, and hormonal systems, amongst others [3]. Moreover, nutrient supply increases observed in fetuses whose pregnancy was characterized by high-fat/high-sugar diets resulted in a need for greater intakes of energy postnatally. This predisposes them to rapid weight gain and permanent changes in their leptin and insulin signal pathways [3,24,25,26]. Systemic inflammation and activation of pro-inflammatory cytokines, such as interleukin 6 (IL6) and tumor necrosis factor (TNF) alpha, is also a matter of high importance that needs monitoring, as it affects early fetal organ (liver, adipose tissue, brain, skeletal muscle, and pancreas) development, thereby precipitating metabolic disorders [27,28].

Moreover, maternal obesity has adverse consequences for both the fetus and the neonate, where overnutrition might cause excessive fetal growth, resulting in macrosomia. A infant being large for gestational age (LGA) results from increased fetal insulin production as a response to maternal hyperglycemia [3]. For macrosomic infants, the short-term consequences might be birth trauma and hypoglycemia. Once a child is born and the supply of maternal glucose is cut off, the infant may continue to produce high levels of insulin but may not have enough glucose available, which can lead to hypoglycemia. At the same time, the long-term consequences of LGA include a higher risk of obesity and type 2 diabetes (T2D) development later in life [3,4,6]. On the other hand, maternal obesity is related to an increased risk of preterm birth, which might result in lower birth weight. Following birth, many small for gestational age (SGA) children may experience “catch-up growth,” which increases the risk of metabolic syndrome. Therefore, infants with low birth weight later in their lives could develop obesity, hypertension, insulin resistance, T2D, and coronary heart disease [29].

Gestational diabetes mellitus (GDM), which develops in the second half of pregnancy, has an incidence rate of 2–16%, depending on ethnic origin [30]. Although pregnancy itself is widely considered a diabetogenic state in nature, the dysfunctional regulation of insulin fetuses, as well as other metabolic regulatory hormones in patients with diabetes, endangers the patient and fetus as opposed to patients who can properly regulate these hormones [6]. Women with uncontrolled diabetes have a greater risk of stillbirth, perinatal mortality, and congenital malformations than those with well-controlled disease [6,31]. A meta-analysis found about a 10% prevalence in macrosomia, with a more pronounced neonatal increase in risk of complications than those without macrosomia. Complications such as shoulder dystocia, OPI, and fractures in pregnancy were observed, in which 6-to-11-fold and 10-to-20-fold increases were recorded [32]. In another meta-analysis, fetal cardiac hypertrophy was observed in 21 out of 26 studies comparing maternal GDM outcomes to non-GDM, while 22 out of 28 were found to have diastolic dysfunction [33]. Maternal metabolic status of increased insulin resistance, leptin, and dyslipidemia has been shown to cross the placental barrier and disrupt fetal metabolism, leading to potential disorders in adulthood [28]. Finally, the mother’s T2D is strongly associated with obesity and T2D development in the offspring. At the same time, maternal type 1 diabetes (T1D) increases the risk of this condition and other autoimmune disorders in children later in their lives [34]. Accelerated maternal risk of complications such as pre-eclampsia, retinopathy, nephropathy, and diabetic ketoacidosis (DKA) has also been observed in patients; therefore, multidisciplinary monitoring as well as other lifestyle changes are indicated in these patients [31].

## 3. Polycystic Ovarian Syndrome

Polycystic ovarian syndrome (PCOS), which is characterized by the presence of polycystic ovaries, hyperandrogenism, and dysfunction in ovulation, is prevalent in ~5–20% of reproductive-aged women worldwide [35]. In PCOS, the dysregulated pattern of gonadotropin-releasing hormone (GnRH), high luteinizing hormone levels, and low follicle-stimulating hormone levels disrupt normal ovarian function. Additionally, anti-Mullerian hormone (AMH), which impairs follicular development, leading to follicular arrest, a hallmark of PCOS, is an important pathophysiological factor [36].

Features characteristic of PCOS, such as hyperandrogenism, obesity, insulin resistance, and metabolic abnormalities, may contribute to the increased risk of obstetric and neonatal complications [37]. The pathophysiological effects are not well documented but may be due to the interplay of estrone, hyperinsulinemia, and subsequent diabetic or hypertensive predispositions [38,39].

Maternal PCOS status may negatively influence infant and childhood growth, cardiometabolic health, reproductive health, and neurodevelopment [40]. While insulin resistance in pregnancy occurs to protect the fetus in pregnancy with PCOS, it may cause a pathologic alteration, which could induce the overexpression of metabolic pathways and cause subsequent adverse pregnancy, fetal, and neonatal outcomes [39]. The consequence of PCOS on reproductive health means that pregnant women with PCOS are more likely to experience complications such as spontaneous abortion, GDM, and preterm birth [9].

Recent findings indicated a potential association between the development of autism spectrum disorder (ASD) and attention-deficit/hyperactivity disorder (ADHD) in those born to mothers with PCOS due to neurodevelopmental effects in gestation [40]. The origins of ASD and ADHD result from multifactorial causes, mainly assumed to be an interplay of genetic and environmental influences [41]. Current studies put forward a hypothesis that exposure to androgens during prenatal development may play a role in the onset of ASD or ADHD [42,43,44,45].

A nationwide Swedish cohort study utilizing register data was carried out to distinguish the effects of prenatal androgen exposure from familial confounding in the relationship between maternal PCOS and the development of ADHD and ASD in offspring. The cohort comprised PCOS-exposed offspring (*n* = 21,280), which were compared with unrelated PCOS-unexposed offspring (*n* = 200,816) and PCOS-unexposed cousins (*n* = 17,295). In this study, children born to mothers with PCOS were found to have an increased likelihood of being diagnosed with ADHD or ASD compared to offspring born to mothers from the general population without PCOS. For ADHD and ASD, comparing affected children to their PCOS-unexposed cousins reduced the observed associations in boys but heightened the estimates in girls. This finding supports to the hypothesis that prenatal androgen exposure may play a role in the development of certain neuropsychiatric conditions, particularly in females, beyond shared familial influences [42]. Comparable results were observed in a nationwide cohort study conducted in Finland, involving 1,097,753 births from 1996 to 2014, with follow-up extending until 31 December 2018, allowing for monitoring until participants reached 22 years of age. National registries were used to link data on the births included and their mothers. Subsequently, data were analyzed from 24,682 children (2.2%) whose mothers had PCOS and compared to 1,073,071 children (97.8%) born to mothers without PCOS. This study demonstrated that maternal PCOS was associated with increased risk for ADHD and ASD in offspring. However, it was also found that children born to mothers with PCOS had a higher risk of other psychiatric problems, among others, sleeping disorders, intellectual disabilities, tic disorders, specific developmental disorders, eating disorders, anxiety disorders, mood disorders, and other behavioral and emotional disorders. Furthermore, when this analysis was stratified based on maternal BMI, it revealed that the risk of any neuropsychiatric disorder was higher in children born to mothers with PCOS who had a normal weight. It was significantly greater in the offspring of mothers with PCOS who were severely obese compared to those born to normal-weight mothers without PCOS [43].

The management of obesity and insulin resistance and the use of pharmaceutical interventions, including metformin prior to pregnancy and during pregnancy, may influence the offspring of mothers with PCOS. Metformin has been used off-label for women with PCOS and/or obesity to enhance pregnancy outcomes. However, the evidence remains equivocal regarding its efficacy in optimizing fertility and pregnancy outcomes [46,47].

In February 2022, European health authorities approved Glucophage^®^ as the first oral medication for diabetes that can be used throughout pregnancy, from conception to birth [48]. This indication follows results from a register-based cohort safety study from Finland, investigating the follow-up of children from over ~4000 pregnancies with metformin for up to 11 years. The cohort included children born between 2004 and 2016 with maternal pregnancy use of metformin or insulin: metformin only (*n* = 3967), insulin only (*n* = 5273), and combined regimen (metformin and insulin; *n* = 889). This study showed that exposure to metformin alone or combined with other treatments, compared to insulin, was not linked to a higher risk of long-term outcomes, including obesity and PCOS, in the offspring. Regarding secondary outcomes, metformin was associated with a higher risk of SGA, while the combined therapy was associated with increased risks of LGA, preterm birth, and hypoglycemia. There was no observed increase in neonatal mortality, hyperglycemia, or significant congenital anomalies [49].

Clinical guidelines differ in their recommendations, and currently, there is no global agreement on the primary pharmacological therapy for GDM. The American Diabetes Association (ADA) [50] and the American College of Obstetricians and Gynecologists (ACOG) [51] both advise insulin as the first-line medication for GDM. Conversely, the Society of Maternal-Fetal Medicine (SMFM) [52] and the National Institute for Clinical Excellence (NICE) [53] recommend the use of metformin for managing GDM. However, to date, there is no international consensus on using metformin during pregnancy in maternal PCOS without diabetes.

The main concern with using metformin during pregnancy is that the drug crosses the placenta, and there is evidence indicating that comparable levels are present in both fetal and maternal circulations [54]. Therefore, using metformin may potentially impact both the developing fetus and the placenta. A recent meta-analysis encompassing 11 studies found that metformin use during the first trimester did not appear to elevate the risk of significant congenital anomalies in children born to women with pregestational diabetes or PCOS [55]. Although no teratogenic effects have been detected in offspring following maternal metformin treatment, growing concerns remain regarding its possible influence on fetal metabolic programming, which could potentially affect the long-term cardiometabolic health of the offspring [47]. Regarding other fetal outcomes following maternal use of metformin, the birth anthropometrics of infants also might be influenced during such therapy. A Nordic randomized controlled trial investigating the effects of metformin in pregnant women with PCOS showed that newborns in the PCOS-metformin group had larger head circumference measurements than infants from the control group. Additionally, this effect was more significant in obese women and individuals with a hyperandrogenic phenotype of PCOS [56]. Regarding the long-term effects of maternal metformin use, a recent meta-analysis comprising 28 studies and 3976 patients with GDM was conducted to evaluate neonatal, infant, and childhood growth after pregnancy. This analysis demonstrated that neonates whose mothers took metformin throughout the gestation period had significantly lower birthweight and anthropometrics than those who were exposed to insulin. While the risk of macrosomia was decreased by 40% in the group treated with metformin, however, by age 2, infants with a history of maternal metformin use weighed more than those exposed to insulin, and between ages 5 and 9, children from the metformin group exhibited a higher BMI [57].

When it comes to maternal pregnancy outcomes, a systematic review and meta-analysis of 35 studies (*n* = 8033 pregnancies) was conducted, where the indications for metformin treatment were hyperglycemia, obesity, PCOS, or pregestational insulin resistance. This analysis demonstrated that women using metformin had lower gestational weight gain and a modestly reduced risk of pre-eclampsia compared with any other treatment [58]. Similar positive results were produced in another meta-analysis of 12 studies (*n* = 1708 women) investigating the influence of preconception and first-trimester metformin treatment on pregnancy outcomes among women diagnosed with PCOS. The main result of the meta-analysis was the miscarriage rate, which was defined in this study as the loss of pregnancy before 20 completed weeks of gestation. Secondary outcomes included the clinical pregnancy rate and the live birth rate. Meta-analyses showed that metformin use in women with PCOS during pregnancy may be beneficial, particularly in improving clinical pregnancy rates, reducing miscarriage, and increasing live birth [59].

In addition to its endocrine effect, transmission of PCOS is also possible. Transgenerational transmission of PCOS from mother to offspring via epigenetic mechanisms is an area under active research. Recent laboratory research demonstrates that late gestation exposure to AMH resulted in transgenerational transmission of PCOS-like phenotype in mice. Encouraging pre-clinical studies have demonstrated the reversal of some PCOS-like phenotypic traits using epigenetics-based therapy when applied to laboratory mice [60]. These findings point to a compelling research question: Could epigenetic therapies reduce the transmission of PCOS characteristics in future generations?

Women living with PCOS are denoted as high-risk pregnancies that require frequent and timely antenatal care. Lifestyle modifications, dietary amendments, and metformin use during the pregnancy of a mother with PCOS may potentially improve some fetal and maternal outcomes [40,46]. Future research considering epigenetic therapies, as well as greater clarity on metformin use during pregnancy, will be valuable.

## 4. Thyroid Disease

Pregnancy itself increases the risk of developing thyroid dysfunction and disease, including hypo- and hyperthyroidism, which potentially can lead to maternal and fetal adverse outcomes. Currently, thyroid disease has become one of the most common disorders in the world, and women have a greater risk of developing thyroid pathology than men. One in eight women will have thyroid disease during her lifespan [61]. Hypothyroidism affects 3–5% of all pregnant mothers, making it the most prevalent thyroid disorder linked to pregnancy, and is associated with negative outcomes for both pregnancy and newborn [62]. The most common reason for hypothyroidism in pregnant women is the autoimmune disease called Hashimoto’s thyroiditis (HT). It can be a pre-existing issue or an initial presentation of the disease [63,64]. During pregnancy, several changes occur in the levels of thyroid hormones due to the physiological adaptations that support both the mother and the developing fetus. Two main hormones can influence thyroid function during pregnancy: hCG and estrogen. In the first trimester, thyroid-stimulating hormone (TSH) levels may decrease due to the stimulating effects of hCG. However, its level typically returns to normal or maybe slightly elevated in the later trimesters as pregnancy progresses. Estrogen elevates the concentration of thyroxine-binding globulin (TBG), which results in a rise of the total thyroid hormone levels with an unchanged level of the “free” hormone [64]. Finally, throughout pregnancy, iodine requirements increase, and any insufficient iodine intake may lead to hypothyroidism, since iodine is crucial for the production of thyroid hormones [62]. The risk profile of maternal hypothyroidism includes an increased likelihood of preterm birth, pre-eclampsia, placental accidents, stillbirth, and miscarriage [7]. Moreover, untreated or inadequately treated hypothyroidism in a mother has several negative impacts on the fetus. Over the first months of pregnancy, the fetus entirely relies on the mother’s production of thyroid hormone. As pregnancy progresses, the fetus starts to secrete its own thyroid hormones; however, it still depends on the mother’s sufficient intake of iodine. Thyroid hormones are essential in the brain development of the fetus, and deprivation of the maternal thyroid hormone due to hypothyroidism can lead to neuropsychological disorders in the offspring. It is well documented that maternal hypothyroidism is related to lower Intelligence Quotient (IQ) and impaired psychomotor development of offspring [65]. Finally, maternal overt hypothyroidism increases many obstetrical risks, including gestational hypertension, low birth weight, and premature delivery. Treatment, especially early in pregnancy, helps to mitigate these risks [66].

When hyperthyroidism occurs during pregnancy, it is usually caused by an autoimmune disorder called Graves’ disease. The primary feature of Graves’ disease is the production of autoantibodies, specifically the TSH receptor antibodies (TRAbs), which mimic the action of TSH. TRAb antibodies bind to the TSH receptor on thyroid follicular cells and lead to continuous stimulation of the thyroid gland, resulting in increased synthesis and secretion of thyroid hormones. Throughout pregnancy, TRAbs produced in the course of a mother with Graves’ disease may cross the placenta and affect the developing fetus’s thyroid. In the case that offspring is born with these antibodies, a child might develop neonatal Graves’ disease. The first signs of hyperthyroidism commonly occur within the first 2 weeks of life. Fortunately, neonatal hyperthyroidism caused by these antibodies lasts only a few weeks until the mother’s antibodies disappear [67].

Overt hyperthyroidism in pregnancy, especially secondary to Graves’ disease, is an indication for treatment, given the risks to the fetus and mother. However, treatment itself can have its fetal effects. Thionamide antithyroid drugs (ATDs), including methimazole and propylthiouracil (PTU), are the first line of treatment for Graves’ disease in pregnancy. However, they cross the placenta [68]. Additionally, maternal hyperthyroidism is associated with an elevated risk of pre-eclampsia, preterm birth, fetal thyroid abnormalities (both hyper- and hypothyroidism), fetal growth restriction, and fetal goiter [7]. In utero exposure to autoimmune thyroid disease has also been reported to increase the risk of childhood acute lymphoblastic leukemia (ALL) in offspring [7].

The management of pregnant patients with thyroid disease involves several contentious issues, including transitioning to non-teratogenic medications, dosage recommendations, acceptable thyroid hormone ranges, and the timing of follow-up appointments. Current practice emphasizes individualized treatment plans to ensure optimal care and promote a healthy pregnancy [69]. Monitoring maternal thyroid values and pharmaceutical treatment in the setting of both hyper- and hypothyroidism is critical in ensuring proper fetal development and optimizing overall pregnancy health [7]. The goal is to maintain thyroid hormone levels pre- and postnatally to minimize the documented risks to the fetus. Close observation during pregnancy of women with hyperthyroidism, adherence to medication regimen, and regular laboratory tests are essential measures in maintaining optimal thyroid hormone levels to reduce the risk of complications to the developing fetus.

## 5. Cushing’s Syndrome

Cushing’s syndrome (CS) presents as systemic manifestations caused by long-term exposure to glucocorticoids, which can arise from different causes and involve disturbances in the normal daily rhythm of cortisol secretion. Being a rare syndrome, its annual incidence is estimated to be 40–70 per million yearly in the general population, occurring mainly in adults, particularly women, usually aged 30 to 50 years [70]. CS during pregnancy is extremely uncommon because hypercortisolism can cause hypogonadotropic hypogonadism, disrupt ovulation, and ultimately result in infertility. Since the first case reported in 1953 by Hunt and McConahey, there are still small numbers of patients with CS during pregnancy recorded in the literature [71,72,73,74]. When CS presents in pregnancy, it is predominantly caused by an adrenal adenoma, accounting for approximately 40–60% of cases [75]. Diagnosing CS secondary to pregnancy is challenging because gestation is related to physiological changes that lead to notable alterations in endocrine hormone levels. The placenta, which acts as an additional endocrine gland in pregnancy, secretes corticotropin-releasing hormone (CRH), adrenocorticotropic hormone (ACTH), and cortisol, particularly in the first trimester. Hormones produced by the placenta impact the activity of the maternal endocrine glands, including the mother’s hypothalamic–pituitary–adrenal (HPA) axis [76,77,78]. Those changes in hormone profiles throughout gestation cause sustained physiological hypercortisolism. Then, the coexistence of symptomatology and the physiological effects of pregnancy represent a diagnostic challenge. Frequently, proper diagnosis is delayed because CS symptoms often mimic normal pregnancy-related changes, including hypertension, elevated blood glucose concentration, weight gain, striae, and mood swings [79].

CS in pregnancy is associated with important maternal–fetal morbidity and mortality [10]. Maternal and fetal complications are broad, and those with active diseases carry the most significant risk [72]. Complications encompass stillbirth, premature birth, spontaneous abortion, neonatal death, fetal intrauterine growth restriction, fetal malformation, and adrenal insufficiency [12,73].

CS predisposes to GDM in pregnant women and carries a risk to the fetus. Up to 60% of mothers might have impaired glucose tolerance, which can lead to perinatal complications [11]. The risk of fetal adverse events is increased in a mother with CS in pregnancy treated medically, surgically, or not at all. Such complications include fetal death, pre-term delivery, intrauterine growth restriction (IUGR), fetal and respiratory distress, low birth weight, and fetal hypoadrenalism [72,74,80,81]. Active disease is associated with multiple insults to both maternal and fetal health, and those cured may normalize the risk of maternal–fetal complications [72]. Hence, management of pregnant women with CS requires careful monitoring of cortisol levels during pregnancy to reduce fetal compromise, particularly if adrenalectomy is needed. Close observation of the maternal and neonate HPA axis should be continued after delivery due to the associated risks. In a similar vein, preconception, counseling, and contraception should be recommended to women of childbearing age [72].

## 6. Endocrine Disruptors

Endocrine disruptors (EDs) are environmental chemicals that interfere with normal hormone activity, causing harmful effects in an intact organism, its progeny, or (sub)populations [82,83]. EDs display a non-linear dose–response relationship in the human body. This means that even at low doses of those chemicals carry significant health risks, leading to complex biological responses [84]. Generally, EDs encompass a wide range of natural and manufactured substances, present in products such as pesticides, metals, plastic bottles, food packaging, cleaning agents, flame retardants, toys, and cosmetic items. The main EDs include bisphenol A (BPA), polyfluoroalkyl substances (PFASs), phthalates (PAEs), phenols, polychlorinated biphenyls (PCBs), metals, organochlorine pesticides, and dioxins [84,85]. People can be exposed to EDs via skin contact, orally (ingestion), or inhalation, leading to a range of health issues such as reproductive disorders, neurodevelopmental problems, metabolic syndromes, and certain cancers [86]. Maternal exposure to endocrine-disrupting chemicals (EDCs) during pregnancy can impact maternal health and fetal development. EDs can pass through the placental barrier via umbilical cord blood, reaching fetal circulation and accumulating in fetal tissue, impairing placental development, growth, and function. The placenta is crucial in producing hormones like estrogen, hCG, and progesterone, which are vital for a healthy pregnancy. It is particularly sensitive to EDs due to its high density of hormone receptors. EDs can disturb the hormonal balance by attaching to these receptors and hormone transport proteins or affecting endogenous hormone degradation and synthesis. Moreover, neither the placenta nor the fetus has any protective mechanisms against EDs, making the fetus more susceptible to any changes that could disturb the balance of the fetoplacental environment. Placental disruption can lead to adverse fetal outcomes such as miscarriage, pree-clampsia, preterm birth, SGA, and low birth weight [8,86,87]. Leclerc et. al. showed a link between pre-eclampsia and a high accumulation of BPA in the placenta. This study tested 58 pregnant women (35 with normal blood pressure and 23 in a preeclamptic state) for BPA concentration using a highly sensitive gas chromatography–mass spectrometry (GC-MS) method. The level of BPA was detected in maternal blood, fetal blood, and placental tissue. The results of this study demonstrated that participants with pre-eclampsia had a significantly higher level of BPA in their placentas compared to normotensive individuals [88]. A similar observation was provided by a case–control study of preterm birth conducted in Boston. The study included 50 cases of pre-eclampsia, with urine samples collected at multiple points throughout pregnancy to measure the levels of BPA and nine phthalate metabolites. Again, the findings from this study confirmed that environmental toxicants such as BPA and phthalate metabolites significantly increase the risk of pre-eclampsia during pregnancy [89].

Exposure to these chemicals at an early stage of pregnancy has been associated with disruptions in syncytialization, trophoblast invasion, and spiral artery remodeling. These alterations can cause placental insufficiency, which may result in fetal growth restriction (FGR) due to ongoing hypoxia and inadequate nutrient supply. Recently, Hong et. al. conducted a prospective cohort study aimed at investigating fetal–maternal exposure to EDs during the COVID-19 pandemic and exploring the relationship between EDCs exposure and obstetric complications. The study included 146 mother–child pairs with singleton pregnancies, each providing sufficient maternal urine and cord blood samples for EDC analysis and complete fetal growth data recorded at the time of admission. Findings from this study showed that neonates with asymmetric FGR had significantly higher maternal and fetal BPA levels than those from the normal growth group [90].

Nonetheless, the potentially harmful effects of EDs are not limited to pregnancy alone, as subtle alterations associated with these chemicals may significantly influence developing tissues, thereby potentially changing the developmental course of the new life. EDs also affect epigenetic regulation, leading to inherited alterations in gene expression that do not involve changes to the underlying DNA sequence. Key epigenetic mechanisms influenced by EDs are DNA methylation, histone modifications, and noncoding RNA expression, which play crucial roles in proper organ formation and embryonic development [91].

EDs can alter DNA methylation patterns by interfering with DNA methyltransferases (DNMTs), the enzymes responsible for adding methyl groups to DNA. These epigenetic modifications can lead to hypermethylation or hypomethylation, affecting gene expression and activity. Such changes may silence essential developmental genes or aberrantly activate others, disrupting processes like cellular differentiation and organ formation. EDs can also influence histone acetylation, methylation, and other post-translational modifications. These changes alter chromatin accessibility, impacting transcriptional regulation of key developmental genes. Disrupted histone modification patterns may impair cell differentiation and tissue formation. Finally, EDs may modulate the expression of microRNAs, small non-coding RNAs that regulate gene expression post-transcriptionally. Such alterations can affect multiple developmental pathways, potentially leading to developmental delays or malformations [84,91,92].

Changes in metabolism-related gene expression can contribute to obesity, insulin resistance, T2D, and metabolic syndrome. Recent studies highlighted that EDs, particularly BPA, exhibit obesogenic activity. Once BPA enters the body, it can bind to various hormonal receptors, including estrogen receptors (ER-alpha and ER-beta), androgen receptors, thyroid receptors, and progesterone receptors, as well as other receptors like the peroxisome proliferator-activated receptor (PPAR-y), retinoid X receptor (RXR), toll-like receptors (TLRs), and NOD-like receptors (NLRs) [93,94]. Remarkably, exposure to BPA affects PPAR-y, a key regulator of lipid metabolism and the differentiation of adipocyte tissue. This interaction allows BPA to stimulate pre-adipocyte differentiation into mature adipocytes, thereby contributing to accelerated adipogenesis and greater fat mass accumulation. Moreover, BPA’s interplay with estrogen receptors can disrupt the signaling pathways of leptin and adiponectin, hormones critical in regulating energy balance and insulin sensitivity. Such interference can exacerbate metabolic disturbances and increase the risk of weight gain and obesity [95,96,97].

A systematic review conducted by Chunxue Yang et al. indicated that early exposure to BPA is linked to a higher likelihood of obesity in adulthood, with findings supported by both epidemiological and laboratory research [98]. A cross-sectional study by Liu et al., including children aged 6–17, showed that BPA was related to a higher prevalence of general and abdominal obesity [99]. Similar findings were presented by Mustieles et al. in their cohort study that comprised 298 children aged 9 to 11. In this study, BPA exposure was associated with an increased risk of obesity, especially abdominal obesity, during the peripubertal period [100]. Moreover, BPA can elevate the risk of obesity even at low exposure levels [84].

In particular, it is essential to understand the multiple effects of BPA exposure and other risk factors, including maternal obesity, on increasing the risk of adverse health outcomes in the offspring. Both maternal obesity and BPA exposure during pregnancy are linked to fetal programming, potentially leading to childhood obesity and other metabolic issues later in their children’s lives [101]. These effects are believed to result from multiple mechanisms such as epigenetic modifications, nutrient transfer changes, hormonal regulation disturbances, and impaired glucose metabolism [24,25,26,27,28]. Moreover, external factors like lifestyle choices, dietary habits, and parallel exposure to other chemicals can markedly increase the obesogenic effect of BPA. For example, diets rich in fats and sugars seem to boost BPA’s influence on weight gain by promoting increased fat accumulation and insulin resistance, as BPA can interfere with metabolic tissues like natural estrogens, even at low levels. Additionally, a sedentary lifestyle with insufficient physical activity can further amplify BPA’s obesogenic effects by impairing metabolic regulation. These external influences can synergistically intensify BPA’s potential to contribute to obesity [102].

For these reasons, public health messaging and advertisements about the potential consequences of consuming products with BPA in the packaging during pregnancy are essential to raising awareness of the possible implications of using these products during pregnancy. The National Institute of Health (NIH) has formulated guidelines to reduce dietary exposure to EDCs based on food selection, cooking, and storage. Examples include purchasing seasonal produce, reducing canned fish or frozen seafood consumption, and avoiding heating foods encased in plastic [8]. Adherence to these guidelines is essential to protect and optimize fetal health. Maintaining a healthy lifestyle with proper nutrition alongside implementing measures to reduce or eliminate EDC exposure during pregnancy is crucial. The urgent need for dose–response research, biomonitoring of EDCs, and policy actions to reduce exposure has been documented [103].

## 7. Conclusions

In conclusion, understanding the role of maternal endocrinopathies and endocrine disruptors on maternal–fetal health is essential and demands a multifaceted, multidisciplinary, and individualized approach to ensure a safe pregnancy. These challenges, which are not limited to hormonal imbalances (e.g., diabetes, PCOS, thyroid disorders), result in life-altering maternal complications such as GDM, preterm labor, pre-eclampsia, and spontaneous abortions while also impacting the health of the fetus (congenital malformations, perinatal mortality, and growth restrictions). The widespread presence of endocrine disruptors (e.g., BPA, phthalates) and other environmental factors further exacerbates these risks.

Numerous studies, as observed in this review, establish associations between maternal–fetal endocrinopathies, endocrine disruptors, and adverse outcomes; however, our understanding of the mechanisms of these associations remains fragmented. The concurrent relationship between endocrinopathies and EDCs is also not as thoroughly investigated. This query refers more to their exacerbating and compounding effects, as well as how they mitigate one another, because treating them as separate entities limits our holistic and collective understanding of these factors.

Despite the growing body of literature on EDCs, there are several limitations hindering the current research body. Some of these include, but are not limited to, small sample sizes and the reliance of many studies on inadequate modes of data collection, such as self-reports, while other studies suffer from the limitations of unstandardized and thereby insufficient biomarker assessments. All of these limitations may tamper with the integrity of the data collected. Additionally, there is a need for region-specific data, as many regions involve environments, diets, and regulations that might not necessarily be found in others, which also impact the generalization of findings to the broader population.

Although this narrative review provides valuable insight and intervention on the complex interactions of maternal endocrinopathies and fetal development, it should be interpreted alongside systematic reviews. Its broad selection criteria and lack of reproducibility are key limitations that impact the reliability and quality of the research. This review offers a conceptual understanding of key information on the topic; a more comprehensive analysis of the literature is required to support a deeper understanding of the literature.

Further studies should focus on longitudinal designs and outcomes over time while also implementing data collection and assessing methods that are standardized and able to be adequately regulated.

Table 1 provides an overview of maternal endocrinopathies and their environmental impact on fetal health. Effective management of these obstacles includes proactive strategies involving preconception counseling/screening, identification, and reduction in exposure to endocrine disruptors, as well as education, awareness, and pharmacological management. Figure 1 is an infographic summarizing the necessary interventions throughout pregnancy. By proactively addressing the risk factors, healthcare providers can enhance maternal health, improve fetal outcomes, and support safe, full-term pregnancies, promoting long-lasting health for both mother and child.

## 8. Future Directions

The increasing prevalence of maternal endocrinopathies such as diabetes mellitus, thyroid disorders, and adrenal dysfunction, coupled with widespread exposure to endocrine-disrupting chemicals, underscores the urgent need to elucidate their combined and individual impacts on fetal and neonatal health. The current literature highlights associations between these maternal conditions and adverse outcomes. Moreover, research over the past few decades has increasingly documented that exposure to certain environmental chemicals—such as bisphenol A (BPA), phthalates, polychlorinated biphenyls (PCBs), and pesticides—can interfere with hormonal systems during critical windows of fetal development. However, gaps remain regarding underlying mechanisms, long-term effects, and effective intervention strategies. In particular, our knowledge regarding the impact of exposure to endocrine disruptors during pregnancy on the fetus and newborn is still limited and not thoroughly understood. While existing studies have provided valuable insights, there remains a critical need for comprehensive research to fully understand the mechanisms, long-term outcomes, and mitigation strategies associated with prenatal exposure to EDs.

In particular, it is essential to conduct longitudinal cohort studies tracking pregnant women from early gestation through childbirth and into childhood. These studies should aim to correlate specific EDs exposures with developmental milestones, neurobehavioral outcomes, and endocrine health over time. Moreover, mechanistic investigations focusing on the interaction between EDs and hormonal signaling at the molecular and cellular levels could provide more detailed information. By elucidating the specific pathways affected—such as epigenetic modifications, receptor interactions, and gene expression changes—researchers can identify the underlying mechanisms that lead to adverse developmental outcomes.

It might be helpful to develop standardized, sensitive methods for detecting and quantifying EDs levels in biological matrices such as blood, urine, amniotic fluid, and cord blood. This will improve exposure assessment accuracy and facilitate risk stratification. Finally, studies identifying specific gestational periods when the fetus is most susceptible to EDs are fundamental. Understanding these windows can inform guidelines for exposure avoidance and policy development.

Addressing the gaps in understanding the impact of endocrine disruptors during pregnancy requires a multidisciplinary, comprehensive approach. Future research should aim to elucidate mechanisms and outcomes and inform effective prevention, regulation, and clinical management strategies to safeguard maternal and child health.

## 9. Strengths and Limitations

The review paper provides a comprehensive overview of the impacts of maternal endocrinopathies and exposure to endocrine disruptors during pregnancy on fetal and neonatal health, highlighting key mechanisms and potential outcomes. However, limitations include potential biases from selective literature inclusion and a lack of long-term epidemiological data, which may restrict the ability to draw definitive conclusions about causality and the full scope of health consequences. In addition, several EDs may be crucial in impacting maternal health and fetal development. This review paper mainly focused on bisphenol A because it is the most widely documented and representative ED with established relevance. Consequently, other EDs are just briefly mentioned. The presented paper also emphasizes correlations rather than establishing definitive causal relationships, underscoring the need for further longitudinal and experimental research to clarify the precise mechanisms involved.

## Figures and Tables

**Figure 1 biomedicines-13-01965-f001:**
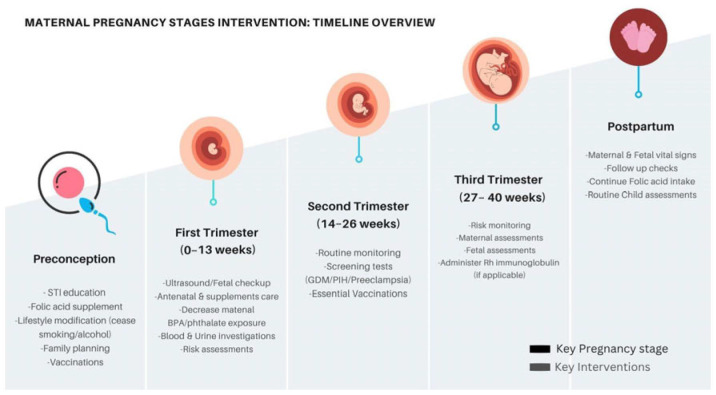
Maternal pregnancy stages and key interventions.

**Table 1 biomedicines-13-01965-t001:** Overview of maternal endocrinopathies and their influence on the fetus.

Endocrinopathy (Study/Sources)	Time of Onset	Effects in Pregnancy	Maternal Complication	Fetal Complication	Management
Obesity Sanli et al. [3], Agarwal et al. [4], Howell et al. [28], Yang et al. [98].	Pre-existing	Increased insulin resistance Metabolic signaling pathway disruptions	Increased risk of GDM Pre-eclampsia Placental dysfunction	LGA Epigenetic modifications Growth restriction	Increase physical activity (at least 30 min/day of moderate exercise). Introduce a balanced diet: a variety of nutrient-rich foods, focusing on fruits, vegetables, whole grains, lean protein, and low-fat or fat-free dairy. Limit intake of saturated and trans fats, sodium, and added sugars.
Diabetes Agarwal et al. [4], Bapayeva et al. [6], Howell et al. [28], Wu et al. [31], NICE [53].	Pre-existing or the 2nd half of pregnancy	Increased insulin resistance	Nephropathy, retinopathy, DKA Pre-eclampsia	Congenital malformations Macrosomia, perinatal death, neonatal hypoglycemia	Adequate glycemic control in the first trimester. Multidisciplinary surveillance and an adequate plan for delivery.
PCOS Pattnaik et al. [9], Yu et al. [39], NICE [53].	Pre-existing	Increased risk of insulin resistance, obesity, and metabolic abnormalities	Higher incidences of spontaneous abortion, preterm birth, and GDM Pregnancy-induced hypertension	Increased risk of perinatal death	Metformin use to improve outcomes. Frequent and timely antenatal care
Thyroid Diseases Maulik et al. [7], Huget-Penner et al. [66], Tsakiridis et al. [69].	Pre-existing	Increased TSH hypothyroidism	Elevated risk of pre-eclampsia	Increased risks of low birth weight, premature delivery, and hypertension Fetal growth restriction and fetal goiter	Transition to non-teratogenic medication and individualized treatment plans. Monitoring thyroid values.
Cushing’s Disease Cholekho et al. [12], Hakami et al. [70], Caimari et al. [72], Luger et al. [73], Bronstein et al. [81].	Pre-existing, very rare in pregnancy	Impaired glucose tolerance, adrenal insufficiency, and hypercortisolism	Spontaneous abortion and premature birth	Fetal growth restrictions, respiratory distress, and fetal death	Parental counseling and close monitoring of cortisol levels.
EDCs (Bisphenol A) Rolfo et al. [8], WHO [82], Toledano et al. [86], Leclerc et al. [88], Cantonwine et al. [89], Hong et al. [90].	Exposure during pregnancy	Hormonal disruption Decrease in micronutrient supply	Increased disposition for pre-eclampsia Preterm birth	Placental dysfunction Developmental abnormalities	Careful diet and planning involving selecting seasonal produce and proper cooking and storage

EDC—endocrine-disrupting chemicals; DKA—diabetic ketoacidosis; GDM—gestational diabetes mellitus; LGA—large for gestational age.

## Data Availability

Not applicable.

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
