# Peer review of "The Effects of Maternal Endocrinopathies and Exposure to Endocrine Disruptors During Pregnancy on the Fetus and Newborn"

_biomedicines, 2025, doi:10.3390/biomedicines13081965_

Round 1
Reviewer 1 Report
Comments and Suggestions for Authors
This is an interesting paper.
However, many data refer to USA and cannot be generalized to other countries. Moreover, PCOS data lack recent studies highlighting AMH's key role in its epigenetic effects.
Page 3, line 100. Prevalence of obesity varies depending on countries and is much lower in many areas. In addition, at least in Mediterranean countries, moderate or severe obesity do not represent 50% of the total number of obese patients.
Epigenetic consequences of PCOS are well demonstrated but the mechanisms may be different from those initially hypothesized. The role of increased AMH seems to be potentially very important. The PCOS section needs to be rewritten.
Author Response
Thank you for the time and effort in reviewing and providing feedback on our manuscript, and we are grateful for the insightful comments and valuable improvements to our paper. We have carefully considered your suggestions and revised the manuscript accordingly. Below, we provide the point-by-point responses.
- This is an interesting paper. However, many data refer to USA and cannot be generalized to other countries. Moreover, PCOS data lack recent studies highlighting AMH's key role in its epigenetic effects.
Answer: Thank you for your important comment. We agree with your suggestions; therefore, some additional references have been added from European and Middle Eastern Journal papers. In addition, the PCOS section has been rewritten, including the role of anti-Mullerian hormone and its involvement in the pathophysiology of PCOS and important future epigenetic considerations relevant to therapy and treatment to prevent neonatal complications. Please see lines 174-176 and 286-298. Also, the lines 234-240 have been edited for more conciseness.
- Page 3, line 100. Prevalence of obesity varies depending on countries and is much lower in many areas. In addition, at least in Mediterranean countries, moderate or severe obesity do not represent 50% of the total number of obese patients.
Answer: Thank you for this valid point. We have removed this statistic (on page 3, line 106) from the manuscript as the epidemiology of obesity worldwide is not central to the objectives of the paper.
- Epigenetic consequences of PCOS are well demonstrated but the mechanisms may be different from those initially hypothesized. The role of increased AMH seems to be potentially very important. The PCOS section needs to be rewritten.
Answer: As suggested, the PCOS section has been rewritten to reflect contemporary epigenetic consequences and the role of anti-Mullerian hormone.
Reviewer 2 Report
Comments and Suggestions for Authors
The author focuses on the current endocrine related diseases and the impact of exposure to endocrine disruptors (EDCs) during pregnancy on fetal development. The author has provided a review of the relevant content, but it seems that some adjustments are needed. 1) The author mainly focuses on the impact of EDCs exposure during pregnancy on development. But the introduction about EDCs appears in the last section. If the author's focus is on the impact of EDCs, can the relevant introduction be moved forward? 2) Should we introduce the structure of epidemiological research animal experiments cellular evidence mechanism exploration for each type of endocrine related disease in pregnant women, and then explain the impact of environmental endocrine disruptors? 3) Should a mechanism diagram be added for each section? Contains EDCs related molecular initiation events, critical timing of disease occurrence, and outcomes. Can this help readers gain a deeper understanding of the author's ideas? 4) The current manuscript seems to be a summary and compilation of the paper. Should the author supplement their own thoughts on relevant research, point out the shortcomings of current research and future research ideas? 5) The author seems to have not provided a detailed introduction to the DOHaD theory, which is currently the core idea of the manuscript. 6) In fact, there are many types of EDCs. If the author only uses BPA as an example, it is recommended to focus on the relevant content of the title and manuscript.
Author Response
We would like to thank the reviewer for the careful and thorough reading of this manuscript and their critical assessment of our work. We have taken the comments on board to improve and clarify the manuscript. In the following, we address their concerns point by point.
The author focuses on the current endocrine-related diseases and the impact of exposure to endocrine disruptors (EDCs) during pregnancy on fetal development. The author has provided a review of the relevant content, but it seems that some adjustments are needed.
- The author mainly focuses on the impact of EDCs exposure during pregnancy on development. But the introduction about EDCs appears in the last section. If the author's focus is on the impact of EDCs, can the relevant introduction be moved forward?
Answer: Thank you for this important feedback. This research contains two areas of focus:
- Endocrinopathies, which include obesity and diabetes mellitus, PCOS, thyroid disease, and Cushing’s syndrome.
- In addition to that, Endocrine disruptors investigate the effect of environmental factors such as nutrition and chemical exposures, e.g., BPA, on the fetus.
While we acknowledge the endocrine-disrupting chemical section is more extensive than the preceding paragraph on Cushing’s Syndrome in the endocrinopathies section, we presented the endocrinopathies first, followed by endocrine disruptors second, to maintain logical progression. We hope you will understand our point of view.
- Should we introduce the structure of epidemiological research animal experiments cellular evidence mechanism exploration for each type of endocrine related disease in pregnant women, and then explain the impact of environmental endocrine disruptors?
Answer: Thank you for your important comment. We agree with this suggestion, so three epidemiological and laboratory papers are included on obesity, the effects of BPA, and the epigenetic effects of PCOS consecutively.
- Should a mechanism diagram be added for each section? Contains EDCs related molecular initiation events, critical timing of disease occurrence, and outcomes. Can this help readers gain a deeper understanding of the author's ideas?
Answer: We appreciate this suggestion. However, the authors believe the summary table and diagram are sufficiently comprehensive to summarize the key findings, as the focus of the research is the outcome and management of endocrinopathies and endocrine-disrupting chemicals on the fetus, which is concluded in the text and corresponding summary table.
- The current manuscript seems to be a summary and compilation of the paper. Should the author supplement their own thoughts on relevant research, point out the shortcomings of current research and future research ideas?
Answer: Thank you for this insightful comment. We understand the value of not only summarizing existing research but also supplementing it with critical analysis. In response, we have included:
- Our own perspective on the implications and findings/how they relate to broader research. Please see lines 533-539.
- A more detailed discussion on the shortcomings and limitations currently in research, such as small sample sizes, limited follow-up up and proper assessment of exposure. Please see lines 540-547.
- Suggestions for future research directions based on identified gaps, particularly in understanding long-term impacts of endocrine disruptors on development, thereby indicating the need for longitudinal studies, region-specific data on data prevalence. Please see lines 555-557.
- The author seems to have not provided a detailed introduction to the DOHaD theory, which is currently the core idea of the manuscript.
Answer: Thank you for this important observation. We agree that the Developmental Origins of Health and Diseases (DOHaD) theory is key in understanding the implications highlighted in this work. In response, we have made changes to the introduction to include the DOHaD theory and its relevance to our research. We hope this provides a clearer context for the framework used in this manuscript. Please see lines (59-67).
- In fact, there are many types of EDCs. If the author only uses BPA as an example, it is recommended to focus on the relevant content of the title and manuscript.
Answer: We thank the Reviewer for this helpful suggestion. We acknowledge that there are many types of EDCs; however, we chose to focus on Bisphenol A in this manuscript as it is the most widely documented and representative EDC with established relevance to our review. We decided to add a paragraph regarding the strengths and limitations of this review, in which we mentioned this issue.
Reviewer 3 Report
Comments and Suggestions for Authors I believe that the manuscript is useful and very well documented, with a major impact on themanagement of pregnant women suffering from the conditions discussed in this Review. However,
I have the following recommendations: 1. Present concrete numbers, for example in percentages, for all maternal conditions that impact
the fetus. 2. I suggest that you shorten very long paragraphs, such as the paragraph between lines 101
and 126. 3. For easier understanding, some graphical representations would be useful.
Author Response
We would like to thank the reviewer for the careful and thorough reading of this manuscript and their critical assessment of our work. We have taken the comments on board to improve and clarify the manuscript. In the following, we address their concerns point by point.
I believe that the manuscript is useful and very well documented, with a major impact on the
management of pregnant women suffering from the conditions discussed in this Review. However,
I have the following recommendations:
- Present concrete numbers, for example, in percentages, for all maternal conditions that impact
the fetus.
Answer: Thank you for pointing this out. We agree that including concrete figures conveys clinical relevance with more clarity and have carefully revised the manuscript to include more necessary data relevant to the study.
The incidence rates of gestational diabetes mellitus, the prevalence of PCOS in women of reproductive age, and the presentation of Cushing’s Syndrome during pregnancy are presented in the main text (lines 148, 171, and 394). Additionally, a recent study from 2023 emphasizes the prevalence of hypothyroid disease, the most common thyroid disorder associated with pregnancy (line 306).
- I suggest that you shorten very long paragraphs, such as the paragraph between lines 101
and 126.
Answer: Thank you for this suggestion. We have carefully revised the manuscript by breaking up and shortening long paragraphs, specifically in this section, to improve flow and readability. We believe these changes better clarify the text. Please refer to the revised section. Please see lines 102-110.
- For easier understanding, some graphical representations would be useful.
Answer: We appreciate this important concern. We understand the importance of visual elements in aiding the reader’s comprehension. In the current version of this review, we have included a summary table and a diagram to support the key findings. We hope these additions address the reviewer’s concern.
Round 2
Reviewer 2 Report
Comments and Suggestions for Authors
No more questions